# The Possible Role of Prescribing Medications, Including Central Nervous System Drugs, in Contributing to Male-Factor Infertility (MFI): Assessment of the Food and Drug Administration (FDA) Pharmacovigilance Database

**DOI:** 10.3390/brainsci13121652

**Published:** 2023-11-29

**Authors:** Sara Baldini, Ahmed Khattak, Paolo Capogrosso, Gabriele Antonini, Federico Dehò, Fabrizio Schifano, Nicolò Schifano

**Affiliations:** 1ASST Sette Laghi—Circolo e Fondazione Macchi Hospital, 21100 Varese, Italy; sbaldini1@studenti.uninsubria.it (S.B.); paolo.capogrosso@gmail.com (P.C.); antoniniurology@gmail.com (G.A.); deho.federico@gmail.com (F.D.); 2Division of Urology, School of Medicine, University of Insubria, 21100 Varese, Italy; 3King’s College Hospital NHS Foundation Trust, London SE5 9RS, UK; akhattak47@gmail.com; 4Antonini Urology, 00185 Rome, Italy; 5Psychopharmacology, Drug Misuse and Novel Psychoactive Substances Research Unit, School of Life and Medical Sciences, University of Hertfordshire, Hatfield AL10 9AB, UK; f.schifano@herts.ac.uk

**Keywords:** pharmacovigilance, male-factor infertility, adverse drug reaction, finasteride, testosterone

## Abstract

Background: A wide range of medications may have a possible role in the development of male-factor infertility (MFI), including various antineoplastic agents, testosterone/anabolic steroids, immunosuppressive drugs/immunomodulators, glucocorticosteroids, non-steroidal anti-inflammatory drugs, opiates, antiandrogenic drugs/5-alpha-reductase inhibitors, various antibiotics, antidepressants, antipsychotics, antiepileptic agents and others. We aimed at investigating this issue from a pharmacovigilance-based perspective. Methods: The Food and Drug Administration (FDA) Adverse Event Reporting System (FAERS) database was queried to identify the drugs associated the most with MFI individual reports. Only those drugs being associated with more than 10 MFI reports were considered for the disproportionality analysis. Proportional Reporting Ratios (PRRs) and their confidence intervals were computed for all the drugs identified in this way in January 2023. Secondary, ‘unmasking’, dataset analyses were carried out as well. Results: Out of the whole database, 955 MFI reports were identified, 408 (42.7%) of which were associated with 20 medications, which had more than 10 reports each. Within this group, finasteride, testosterone, valproate, diethylstilbestrol, mechloretamine, verapamil, lovastatin and nifedipine showed significant levels of actual disproportionate reporting. Out of these, and before unmasking, the highest PRR values were identified for finasteride, diethylstilbestrol and mechloretamine, respectively, with values of 16.0 (12.7–20.3), 14.3 (9.1–22.4) and 58.7 (36.3–95.9). Conclusions: A variety of several medications, a number of which were already supposed to be potentially linked with MFI based on the existing evidence, were associated with significant PRR levels for MFI in this analysis. A number of agents which were previously hypothesized to be associated with MFI were not represented in this analysis, suggesting that drug-induced MFI is likely under-reported to regulatory agencies. Reproductive medicine specialists should put more effort into the detection and reporting of these adverse drug reactions.

## 1. Introduction

The global rate of couple’s infertility is not conclusively defined, although World Health Organization (WHO) figures suggest that up to 48 million couples worldwide are affected with this issue [1,2].

The WHO defines male-factor infertility (MFI) as the failure to achieve a pregnancy after 12 months or more of regular unprotected sexual intercourse in the presence of abnormal semen parameters or abnormal sperm functional assays [3].

The literature over the past decade has suggested an increase in the global prevalence of MFI, marked by a fall in sperm count and seminal fluid volume, with a temporal trend of decline in sperm concentration levels up to 52.4% and a decrease of 59.3% in sperm count from 1973 to 2011 [4].

Even though a number of factors are potentially implicated in its development, 44% of MFI cases are still classified as idiopathic/unexplained [5]. Among the other etiologies associated with MFI, a wide range of medications may have a possible role in its pathogenesis, including various antineoplastic agents, testosterone/anabolic steroids, immunosuppressive drugs/immunomodulators, glucocorticosteroids, non-steroidal anti-inflammatory drugs, opiates, antiandrogenic drugs/5-alpha-reductase inhibitors, various antibiotics, antidepressants, antipsychotics, antiepileptic agents and others. Notwithstanding the recognised impact, which some medications may have in the pathogenesis of MFI, the quality of the evidence in support of drug-induced MFI is relatively low, and the issue remains under-investigated. Though for some molecules, such as chemotherapy agents, there are convincing levels of evidence relating to their association with MFI [6], for other molecules, the quality of the evidence in support of this drug-induced association is either conflicting or relatively low/unsatisfactory, despite the existence of a scientific rationale to better investigate this issue.

Pharmacovigilance is the pharmacological science dealing with the post-marketing detection and analysis of the adverse drug reactions (ADRs) related with an index medication; it is based on the collection of spontaneous reports, as submitted by patients or healthcare practitioners. The Food and Drug Administration (FDA) Adverse Event Reporting System (FAERS) is an international pharmacovigilance database, which contains data relating to the ADR reports submitted to the FDA [7].

We aimed to identify those agents most linked with MFI from a pharmacovigilance perspective and to determine the strength of this association using the FAERS database.

## 2. Materials and Methods

The FAERS database was accessed via the ad hoc online public dashboard. The Medical Dictionary for Regulatory Activities (MedDRA) terms was used to identify those ADRs related to MFI, using the MedDRA term ‘Infertility Male’.

The spontaneously reported ADRs could be submitted on the FAERS database by both patients and healthcare professionals. The data were harvested in January 2023 and comprised all MFI reports submitted in the dataset. Only those drugs associated with more than 10 MFI reports were included for analysis.

Data were obtained from the FAERS Public Dashboard, which is a publicly available web-based tool that allows for the querying of FAERS data related to ADRs reported to the FDA by the pharmaceutical industry, healthcare providers and consumers. The reports submitted to the FAERS comprised a range of parameters, including the following: suspect product active ingredient(s), reason for use, seriousness of the ADR, event date, sex of the patient, patient’s age, patient’s weight, reporter type/reporting source, concomitant product(s) taken by the patient at the same time of the report, country where event occurred and possible literature reference(s) where the event was discussed. The whole database was analyzed, including the reports submitted from 1981 to 2021 (Figure 1).

In order to more accurately assess the strength of the possible association of the identified drugs and MFI, the Proportional Reporting Ratios (PRRs) and their confidence intervals (CIs) were computed for all the drugs identified in this way [8]. The *PRR* was calculated using the following formula:$$PRR=\frac{A}{A+B}/\frac{C}{C+D}$$
where *A* is the number of individual cases associated with the index drug involving ‘MFI’; *B* is the number of individual cases related to the index drug involving any other adverse events; *C* is the number of individual cases involving ‘MFI’ for all the remaining drugs; and *D* is the number of individual cases involving any other ADR associated with the remaining drugs.

Where the PRR is greater than 1, the suggestion is that MFI is disproportionally reported in those taking the index drug, as compared to those who do not. A PRR > 3, along with the lower bound of the CI being > 1, is typically considered as suggestive for a strong signal of disproportionate reporting. Consistent with Capogrosso Sansone et al. [9], a range of secondary, ‘unmasking’, analyses were carried out as well. The identification of candidates masking products still, however, relies on an empirical approach [10]. Hence, to be as conservative as possible, unmasking was obtained in repeating the primary analysis whilst excluding, from the dataset relating to drugs associated with a significant PRR value, all reports in which at least one other drug (e.g., any drug, irrespective of having been suspected of being associated with MFI) was mentioned as a concomitant medication.

Further details were obtained from the analysis of the literature references relating to the reports associated with the drugs showing significant levels of disproportionate reporting. To carry out a quality evaluation about the causality of the suspected drugs analyzed, the Naranjo probability [11] scale was used. Consistent with Gupta and Kumar [12], the scores were independently calculated by two clinicians (e.g., a clinical pharmacologist and a urologist; FS and NS), and possible disagreement issues were discussed.

Statistical analysis was performed using SPSS (Version 24.0. Armonk, NY, USA) software.

## 3. Results

Analysis of the FAERS database revealed 955 reports of MFI. Some 20 medications (Table 1) were associated with 10 or more MFI reports each and were responsible for 408 reports of MFI (e.g., 42.7% of the total number of MFI reports in the whole database) and were included for the analysis.

The PRR values and the related CIs can be found in Table 1, whilst the PRR values obtained after the unmasking analysis are provided in Appendix A, uploaded as Appendix A. Among the medications included for the analysis, finasteride, testosterone, valproate, diethylstilbestrol, mechlorethamine, verapamil, lovastatin and nifedipine were associated with significant levels of disproportionate reporting. Overall, 240 (e.g., 58.8%) reports were made by consumers, while 155 (38.0%) were made by healthcare professionals (Figure 2).

Out of all of the identified reports, 37 (e.g., 9.1%) were associated with a literature reference (e.g., these findings were published as well), 35 of which were reported to the FAERS by a healthcare professional. The countries where the reports were submitted are represented in Figure 3.

Finasteride was associated with a PRR = 16.04 (12.67–20.3; after the unmasking analysis, the PRR value was 12.94 (12.62, 13.27). Finasteride was used in 61 (e.g., 70.9%) reports for alopecia; in 1 report for benign prostatic hyperplasia (BPH); and in the remaining cases, the indication for finasteride prescription was not reported. The mean age associated with the finasteride-related reports, when specified, was 44.2 years. Most (*n* = 49; 57.0%) reports were made by a healthcare professional.

Where specified, testosterone (PRR 3.03 (2.12–4.32); after unmasking: PRR 5.63 (5.54, 5.72)) indication was hypogonadism in 24 cases (e.g., 72.7%), and for 9 reports (e.g., 27.3%), it was used for unknown indications. Most (e.g., *n* = 24; 72.7%) reports were made by a healthcare professional. Patients’ age was specified for 15 reports out of the total, with an average of 43.3 years.

Valproic acid (PRR 1.72 (1.20–2.47); after unmasking: PRR 3.53, (3.46, 3.61)) reports were associated with epilepsy in five (e.g., 15.6%) patients, whilst for six reports (e.g., 18.7%), the medication was prescribed for a psychiatric indication, including affective and psychotic disorders. The mean age of these patients was 31.5 years. Most of these reports (e.g., *n* = 23, 71.9%) were submitted by the consumers themselves.

Regarding diethylstilbestrol (PRR 14.3 (9.13–22.37); after unmasking: PRR 17.98 (17.04, 18.95)), the latest associated report was received by the FDA in 1997. The mean age calculated for these reports was 35.4 years. Out of these reports, 17 (e.g., 85%) of them also included a congenital genitourinary abnormality along with MFI. Diethylstilbestrol reports were submitted by consumers in 19 cases (e.g., 95%).

With respect to verapamil (PRR 1.83 (1.07–3.12); after unmasking PRR 0.79 (0.75, 0.82); n.s.), in most cases, the reason behind this prescription was not specified, whilst in one report, the medication was used to treat hypertrophic cardiomyopathy. The mean age mentioned in these reports was 34.8 years. Out of these, 12 (e.g., 85.7%) reports were submitted by the consumer.

Nifedipine (PRR 1.85 (1.04–3.28); after unmasking: PRR 0.95 (0.91, 0.98); n.s.) indication was specified for two reports (e.g., 16.7%); in both of these reports, the reason for use was hypertension. Patient main age was 37.4 years for these reports. Out of these, 10 reports (e.g., 83.3%) were submitted by the consumer.

In terms of lovastatin (PRR 2.51 (1.44–4.36); after unmasking: PRR 1.08 (1.04, 1.12)), the mean age associated with the related MFI reports was 40 years. Some 12 reports (e.g., 92.3%) were submitted by consumers. The latest FDA report was received in 1996.

No reason for use was indicated for the reports associated with mechlorethamine (PRR 58.71 (36.30–94.94); after unmasking: PRR 0.57, CI: (0.49, 0.66); n.s.) reports. For most (15/29 reports), however, mechlorethamine was identified in concomitance with remaining chemotherapy medications, including vincristine sulfate and/or procarbazine. There were 14 (e.g., 82.3%) mechlorethamine-associated reports submitted by the patients themselves; the patients’ mean age was 33 years.

Finally, according to the Naranjo probability scale, all suspected drugs associated with significant levels of disproportionate reporting scored as having had a ‘possible’ role in causing the MFI ADR.

## 4. Discussion

To the best of our knowledge, this study adds to the existing, albeit for some medications, conflicting or unsatisfactory, evidence of the possible detrimental impact of medications on the development of MFI using a pharmacovigilance approach. Through pharmacovigilance, the safety of the medicines is monitored throughout their use in healthcare practice. The FAERS database, along with the European Medicines Agency (EMA) and the World Health Organization’s Drug Monitoring Program, is considered the international reference standard for ADR reporting [13,14,15,16,17,18].

The MeDDRA term ‘Infertility Male’ was used for the purpose of this study, instead of other MeDDRA terms, which identify mere alterations in the semen parameters (e.g., “oligozoospermia”, “altered seminal parameters” etc.). In fact, a mere alteration in the semen parameters does not necessarily imply MFI, even if one could argue that MFI secondary to medications’ use could more frequently present with altered semen parameters, rather than with a normal semen analysis [19]. Traditional semen analyses are commonly used to determine semen quality but have critical deficiencies, such as poor reproducibility, poor fertility prediction and significant inter-laboratory and intra-individual variability [20]. Semen analysis is an indirect measure of fertility that reports the characteristics of the semen rather than the ability of the sperm to conceive a healthy child [21].

In nearly 30–40% of infertile men, it is not possible to identify a definite cause for their semen-analysis alterations and, therefore, these cases are classified as having idiopathic MFI [22]. Oxidative stress has been identified as one of the main mechanisms by which various endogenous and exogenous factors can lead to this condition [22]. The possible role of drugs in contributing to MFI is a poorly investigated issue, and part of the infertility classified as idiopathic could be related to drug intake. Indeed, in our analysis, we found a relatively small number of FAERS MFI reports, and most of these were reported by consumers rather than by a healthcare professional.

Our analysis highlighted disproportionate reporting levels associated with finasteride, with over 21% of reports having been associated with this agent. Finasteride is a 5 alpha-reductase inhibitor that acts as an anti-androgen [23] by reducing the formation of the more bio-active androgen 5-alpha dihydrotestosterone (DHT) in target tissues, such as the prostate gland and the hair follicles [24]. Its clinical indications include the treatment of benign prostate hyperplasia (BPH) and male pattern hair loss in younger men. The prevention of the transformation of testosterone to its more biologically active metabolite 5α-dihydrotestosterone could theoretically negatively impact spermatogenesis, even though studies have shown conflicting results. A prospective study on 99 patients suggested a detrimental effect of finasteride 5 mg on total sperm count, semen volume, sperm concentration and sperm motility [25]. Conversely, finasteride 1 mg did not seem to affect seminal production, sperm motility or sperm morphology in 181 young healthy men according to the findings of a double-blind placebo-controlled RCT multicenter study [26]. However, this study evaluated only fertile men, and those patients with a history of infertility and with abnormal semen parameters were excluded from the study. Samplaski et al. [27] prospectively collected data relating to 27 patients on finasteride out of 4400 men seeking treatment for infertility, identifying a statistically significant improvement in the sperm counts (e.g., median 11.6 folds) for most patients after finasteride discontinuation. The authors concluded that low-dose finasteride may have only a mild influence on sperm parameters in healthy men, but this effect may be amplified in infertile men. Liu et al. [28] and Chiba et al. [29] documented the reversibility of the seminal parameters’ alteration after the discontinuation of finasteride in their case series. Hence, discontinuing finasteride in oligospermic and azoospermic men of reproductive age is, therefore, typically considered advisable. Most reports of finasteride-associated infertility have studied the effects of finasteride on spermatogenesis in relation to semen parameters, but it has been suggested that this effect may be secondary to an impairment in the genetic integrity of the sperm cells. A number of case reports [30,31] identified a reduction in the spermatic DNA fragmentation index (DFI) after finasteride cessation. This finasteride-associated sperm genotoxicity may exert an unfavorable effect on the fertility potential of the patients exposed to finasteride, which may be secondary to an impairment of the embryo-implantation process. The male reproductive tract may be particularly sensitive to inhibitors of 5α–reductase as there are levels of 5α-reductase activity in all of the tissues of the male reproductive tract, the function of which is likely regulated by androgens [32]. A preclinical study [33] showed that finasteride may cause an impairment in the spermatogenic process due to possible changes in the structure and function of the epididymis, without any significant alterations in sperm production. The sperm cells were found to transit more quickly through the epididymis, which may compromise their maturation, hence resulting in an impairment in their function. Although the above evidence seems to suggest that finasteride may have a detrimental impact on the spermatogenesis of some patients exposed to it, a number of issues of relevance should be raised. First, seminal quality alterations do not necessarily result in MFI. MFI must be considered as a possible manifestation of the so-called post-finasteride syndrome (PFS), a debated clinical entity, which develops during finasteride treatment and may persist after discontinuing it [34]. Additionally, the relatively high number of ADR reports and consequent high PRR levels for finasteride identified in this analysis can be partially explained with the notoriety bias issue related with the PFS [35].

Exogenous testosterone is widely prescribed for the management of hypogonadism, but its usage is known to lead to secondary spermatogenic failure due to the suppression of the hypothalamic–pituitary–gonadal axis through a negative feedback [36,37]. The inhibition of LH release leads to the suppression of intra-testicular testosterone production by Leydig cells, which, in addition to the suppression of FSH, leads to decreased germ cell survival and maturation [37]. Stopping the use of exogenous testosterone typically leads to a reversal of the azoospermia in the majority of men after a median period of 3.7 months [38,39], although some authors suggested that the amount of time needed to completely restore the consequences on spermatogenesis can take up to 3 years [40]. In up to one-third (e.g., 27%) of testosterone-associated reports, the indication for prescription was not disclosed in this analysis (Figure 4). A number of recent epidemiological surveys have shown an increasing trend towards the abuse of anabolic steroids, especially among those aiming to increase muscle mass and strength, as well as improving their physical performance. Hence, one could argue that a number of those reports could theoretically be associated with anabolic–androgenic–steroid abuse [41,42].

The mechanism of action of valproic acid is not fully understood, although its anticonvulsant effect has been attributed to the blockade of voltage-gated sodium channels and increased brain levels of gamma-aminobutyric acid (GABA) [43,44]. In comparison with non-epileptic men, those affected by epilepsy have been reported to have a higher likelihood of fertility issues; this is possibly associated with the sexual dysfunction issues, which have been reported in up to 71% of these patients [45]. Guo et al. [46] prospectively evaluated the sperm quality of 44 young males with epilepsy (e.g., 23 being treated with valproic acid and 21 receiving oxcarbamazepine) and 30 age-matched healthy individuals. The sperm parameters were significantly reduced in those patients on valproic acid vs. both healthy individuals and those on carbamazepine. A number of different mechanisms have been proposed to explain the decreased levels of fertility among valproate users [47]. Animal models [48] have shown that valproate is capable of increasing prolactin levels, which, in turn, reduced LH and FSH levels. A number of human studies confirmed lower circulating levels of LH and FSH in those patients on valproic acid [48,49]. Further effects on sex hormone release have been theorised through the negative GABAergic effect on the gonadotropin-releasing hormone (GnRH), which, again, can reduce LH and FSH release [47]. The gonadal effects of valproate have also been suggested, with rat models highlighting a 50% reduction in testosterone production [50]. Valproate may also be associated with increased oxidative stress levels in the testicles, due to its capability of inhibiting histone deacetylase, resulting in the hyperacetylation of the histones and in an impairment in the histone-to-protamine transition, with protamine contributing to DNA stability in the spermatogonia process [47]. Mitochondrial dysfunction, being secondary to the decrease in carnitine levels, which results in reduced energy production and, thus, reduced sperm motility, is also hypothesised to be related to valproate use. A number of clinical and preclinical studies identified a correlation between abnormal sperm count, motility, morphology, testicular volume and carnitine levels, which correlates with valproate dose and duration [51,52,53].

Diethylstilbestrol is a synthetic nonsteroidal estrogen, which inhibits the hypothalamic–pituitary–gonadal axis through reducing the testicular synthesis of testosterone; therefore, it has been historically used for chemical castration in the management of prostate cancer [54,55]. The primary indication for diethylstilbestrol prescription was, however, both the prevention of miscarriage in pregnant women and the treatment of menopause and estrogen-deficiency symptoms. The administration of this molecule in pregnant women shows the potential to correlate with a variety of structural abnormalities on the progeny, including congenital genitourinary abnormalities in males, such as varicocele, cryptorchidism and testicular hypoplasia, which can lead to infertility [56].

Mechlorethamine is a nitrogen mustard, which has been used in combination with other antineoplastic agents to treat several types of blood cancers, such as Hodgkin disease, chronic leukemias and polycythemia vera [57]. The first chemotherapy regimen widely used to treat advanced Hodgkin’s disease was the MOPP combination [58], which included mechlorethamine along with vincristine, procarbazine and prednisone, as here identified in a number of cases. The adoption of the MOPP regimen was abandoned over the years in favour of different chemotherapy schemes, which proved to be more effective. Currently, mechlorethamine use is limited to the topical treatment of mycosis fungoides [59], and most likely, this has no influence on spermatogenesis; this may be consistent with the decrease in the mechlorethamine-associated MFI reports in the FAERS database throughout the years.

Serotonin-selective receptor inhibitors (SSRIs) are capable of increasing the extracellular level of the neurotransmitter serotonin by limiting its reabsorption into the presynaptic neuron; they are commonly used in the treatment of both depressive and anxiety disorders [60] but also in the management of premature ejaculation [61]. It has been suggested that their possible effects on sperm quality may be confounded by the underlying diagnosis [62]. Various SSRIs have been found to have a spermicidal effect on semen samples from human donors in vitro [63]. Case reports [64,65] of men taking antidepressants referred for infertility evaluation found them to present with abnormal semen parameters that reversed upon medication cessation. In a study [66] comparing the semen quality of men on SSRIs vs. matched-healthy controls, those patients on SSRIs were found to have lower sperm counts, lower motility, worse sperm morphology and increased DNA damage. Paroxetine was also found to induce an increase in sperm DFI despite otherwise normal semen parameters after five weeks of intake [67]. In a recent retrospective study on 8861 men undergoing semen analysis for fertility evaluation, the SSRI exposure was, however, not associated with any differences in semen parameters vs. the non-exposed men [68]. The mechanism explaining the putative influence of SSRIs over sperm quality is unclear, but it has been postulated that the dysregulation of the tryptophan metabolism may disrupt spermatogenesis [65] and that SSRIs may influence the adenosine triphosphate (ATP) synthesis by inhibiting the oxidative phosphorylation in the spermatic mitochondria [63]. A number of preclinical studies [69,70] seemed to suggest that SSRI-related exposure may also affect the vas deferens motility. Although the existing clinical evidence seems to suggest a possible impact of both SSRIs and serotonin and norepinephrine reuptake inhibitors (SNRIs) on spermatogenesis, the PRR associated with these medications clearly failed to reach statistical significance levels here.

Calcium ions are vital second messengers in human physiology, and the use of drugs that interfere with their function may potentially lead to an interference with a number of biological pathways, thus including spermatogenesis. Even though reduced sperm concentration, impaired serum testosterone, FSH and LH levels were associated with verapamil exposure in a number of preclinical studies on rats [71,72], well-designed observational studies and RCTs to define this association in the clinical setting are still lacking. Calcium channels are highly represented in mature sperm cells, and interference with these may potentially lead to an interference with some of the sperm–egg fertilization processes, such as the acrosome reaction [71]. Calcium ions are also needed for the epididymal acquisition of sperm motility in experimental models [73]. Nifedipine and verapamil may also lead to a decrease in the antioxidant activities of catalase and superoxide dismutase, with a subsequent increase in the sperm–lipid–peroxidation levels [74].

Finally, statins have also been hypothesized to interfere with the spermatogenesis process [75] through their possible influence over the metabolism of cholesterol, which represents the main precursor of steroidal hormones. Cholesterol is also a determinant component of the plasma membrane of the sperm cell, where it certainly influences the protective and fusion capacity of the membrane itself [76].

### Limitations

A number of possible limitations need to be considered when interpreting the present findings. Pharmacovigilance cannot prove the causality of association, and clinical studies are always needed to confirm any signal of disproportionate reporting. If the PRRs of some drugs turned out to be non-significant, this could be because the disproportionality analysis was carried out only among the 20 drugs with the most reports rather than on the whole database, and, for this reason, it cannot be excluded that even those with non-significant PRR values may be associated with a disproportionality of reporting for MFI. Following the unmasking exercise, it appeared from here that some signals of disproportionate analysis actually increased in value, whilst others were non-significant, with this effect having been described before [9]. Although the highest masking effects are obtained by products for which the reaction is known or has been extensively reported [10], a conservative approach was taken into account, and any combination drug, irrespective of having been the subject of an MFI concern or not, was eliminated from the related dataset of interest. One could argue that this may have further amplified the differences between the ‘crude’ and ‘unmasked’ PRR values of interest.

It is likely that ADRs were under-reported to regulatory agencies due to the patient perception of whether infertility was associated with the medication use, the clinician awareness or the extent of use of the medication itself [77]. Indeed, an analysis of the medical history of the patient experiencing the suspected ADR would have been of the utmost importance to better understand a number of relevant issues, including whether some specific categories of patients may be more prone to develop MFI as a consequence of medication intake. However, the patients’ health status is not typically mentioned in the pharmacovigilance databases, including the FAERS. Furthermore, due to the limitations of the pharmacovigilance databases, only reports presenting the effects of single drugs, as opposed to drug mixtures containing the index drug of concern, were considered for the analysis. Although the concomitant drugs taken by the patient at the same time of the suspected drug were, whenever possible, disclosed, the PRR values were computed only for the suspected drugs and not for the molecules being identified in the drug combination. On the other hand, only procarbazine and vincristine, both associated with mechloretamine, were identified in more than three instances out of the whole dataset, and with the related unmasking analysis, their confounding/masking role on mechloretamine was possibly better interpreted. One could wonder about the pharmacovigilance consistency of carrying out indeterminate searches (at times referred to as a “fishing expedition”) in order to obtain, as the case may be here, information relating to the association between MFI and an index drug. Conversely, the computation of both the PRR values and their CI levels is still considered the gold-standard strategy to identify any possible signal of disproportionate reporting for any given medication. In addition to that, one could wonder about a possible pharmacovigilance bias here introduced in assessing only those drugs being associated with more than 10 MFI reports each. This was carried out to avoid the over-inclusion of large numbers of mainly anecdotal, and likely not to reach the statistical significance, reports relating to single molecules in this paper. A similar approach has also been implemented in previous pharmacovigilance studies [77]. Although among the many tools identified, the Naranjo’s scale remains the most comprehensive and easy to use for performing causality assessment of ADRs [78], its applicability in the current study was of limited value. In fact, lots of clinical information to optimally compile the scale (e.g., reappearance of the ADR when a placebo was given; presence of toxicity levels of the suspected drug; occurrence of the same ADR reactions with previously administered drugs, etc.) was not made available in the related reports. It is an inherent limitation of spontaneous reporting, however, that firm evidence cannot usually be produced.

## 5. Conclusions

An accurate collection of the drug history is clearly advisable in every male patient seeking medical help for having issues in conceiving. Patients on testosterone must always be counselled on the detrimental effects of these molecules on fertility. Larger-scale clinical prospective studies are needed to fully elucidate the possible association of finasteride with MFI, particularly in the case of low-dose finasteride, as being used in the treatment of male pattern baldness in young patients. Valproic acid, calcium channel blockers and statins also seemed to be associated with disproportionate reporting levels relating to MFI, although the link between these medications and MFI needs to be better investigated in the clinical setting. The impact of medications on the development of MFI may be an overlooked issue, and more efforts should be put into signaling these ADRs to regulatory agencies. Pharmacovigilance may be the best available tool to clarify this possible association.

## Figures and Tables

**Figure 1 brainsci-13-01652-f001:**
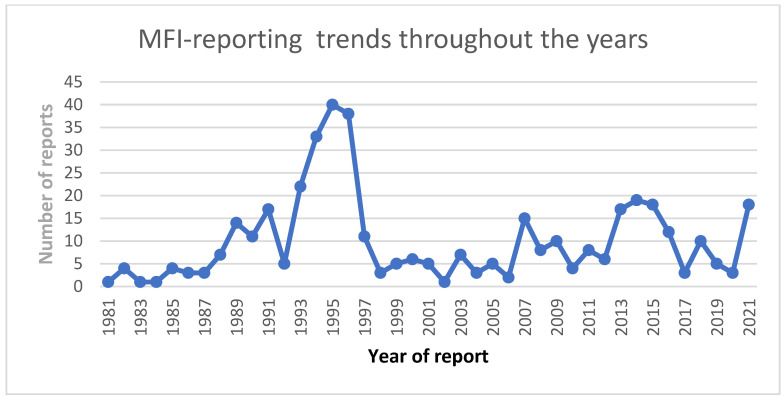
MFI- reporting trends throughout the years.

**Figure 2 brainsci-13-01652-f002:**
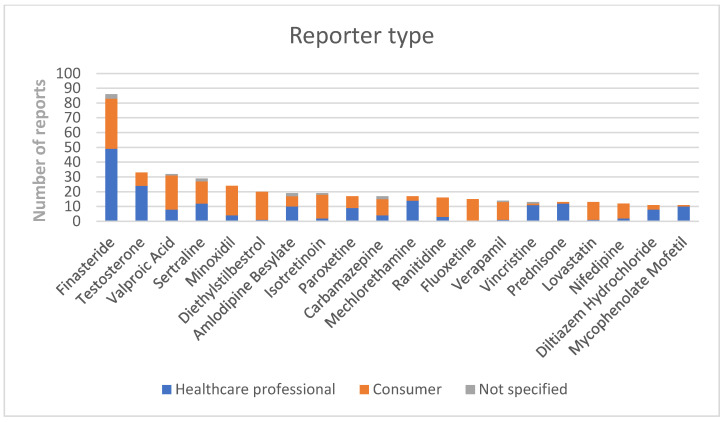
Reporter types.

**Figure 3 brainsci-13-01652-f003:**
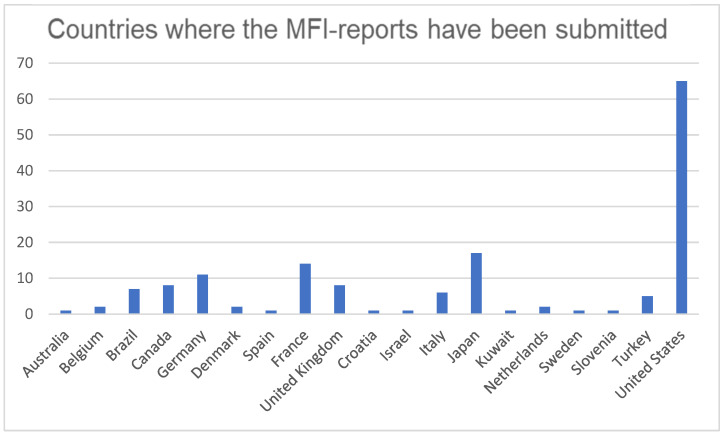
Countries where the MFI-reports have been submitted.

**Figure 4 brainsci-13-01652-f004:**
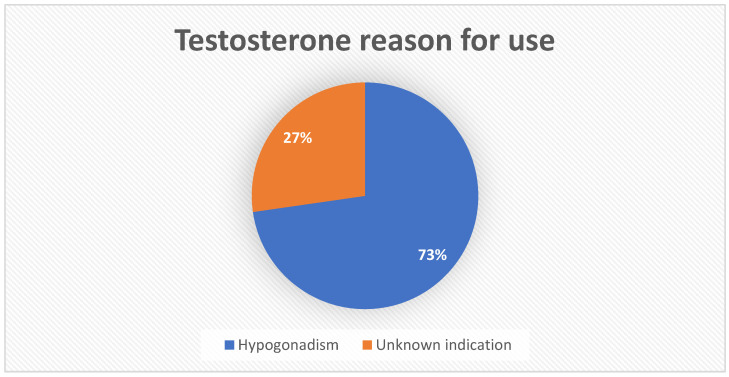
Testosterone reason for use.

**Table 1 brainsci-13-01652-t001:** PRRs and CIs values.

Pharmacological Class	Drug	Number of Individual Cases Associated with MFI in the Database (n)	Percentage of the Individual Cases of MFI When Compared to All the Compounds in the Database (%)	PRR (CI95%) *
5 alpha-reductase inhibitors	Finasteride	86	21.08%	**16.04 (12.67–20.3)**
Steroid hormones	Testosterone	33	8.09%	**3.03 (2.12–4.32)**
Anticonvulsants	Valproic Acid	32	7.42%	**1.72 (1.20–2.47)**
	Carbamazepine	17	4.17%	1.07 (0.66–1.74)
SSRIs	Sertraline	29	7.11%	0.94 (0.64–1.37)
	Paroxetine	17	4.17%	0.92 (0.56–1.5)
	Fluoxetine	15	3.68%	0.51 (0.30–0.85)
Direct Vasodilators	Minoxidil	24	5.88%	0.95 (0.62–1.43)
Nonsteroidal estrogens	Diethylstilbestrol	20	4.90%	**14.3 (9.13–22.37)**
Calcium-channel blockers	Amlodipine Besylate	19	4.66%	0.69 (0.43–1.09)
	Verapamil	14	3.43%	**1.83 (1.07–3.12)**
	Nifedipine	12	2.94%	**1.85 (1.04–3.28)**
	Diltiazem Hydrochloride	11	2.70%	1.49 (0.82–2.72)
Skin and Mucous Membrane Agents	Isotretinoin	19	4.66%	0.84 (0.53–1.33)
Alkylating agents	Mechlorethamine	17	4.17%	**58.71 (36.30–94.94)**
Histamine H2-Antagonists	Ranitidine	16	3.92%	0.10 (0.06–0.16)
Antineoplastic Agents	Vincristine	13	3.19%	1.12 (0.65–1.95)
Glucocorticoids	Prednisone	13	3.19%	0.28 (0.16–0.48)
Statins	Lovastatin	13	3.19%	**2.51 (1.44–4.36)**
Immunosuppressive Agents	Mycophenolate Mofetil	11	2.70%	0.47 (0.26–0.86)

Significant PRR values are marked in bold. * Keys: PRR = Proportional Reporting Ratio. CI95% = 95% Confidence Interval. MFI = male-factor infertility.

## Data Availability

Publicly available datasets were analyzed in this study. This data can be found here: https://www.fda.gov/drugs/questions-and-answers-fdas-adverse-event-reporting-system-faers/fda-adverse-event-reporting-system-faers-public-dashboard.

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
