# Peer review of "The Possible Role of Prescribing Medications, Including Central Nervous System Drugs, in Contributing to Male-Factor Infertility (MFI): Assessment of the Food and Drug Administration (FDA) Pharmacovigilance Database"

_brainsci, 2023, doi:10.3390/brainsci13121652_

Round 1

Reviewer 1 Report

Comments and Suggestions for Authors

Reviewer report

The current study is informative because it informs institutions and healthcare providers on the post-market performance of the drug entity, a parameter not always captured in the clinical studies stage in drug development.

The article is well-written and relevant to the medical practice community. I only have a few comments as follows;

1.       The Method sections

Comment:

Provide a summary description of the database to include how it can be accessed, what the parameters measure (Variables) what is the year span of the data extracted, and so on.. This provides readers with enough information to understand the database.

Comment:

I think there should be some additions to the analysis. The active agent that is associated with MFI needs to be classified pharmacologically stating in which indication they are used, what are they actually used for the case reported, and maybe to which patient group (Male: age distribution)

Also provide time trends and geological location of the reports, who made the reports, what action was taken, and possibly, the general outcome after an intervention. I hope you understand that all these parameters affect the drug therapy.

Consider that this information might be valuable to patient groups that are not medical experts so more information is needed.

2.       The conclusion

Comment on Line 345

This is not in your result kindly review. Perhaps suited for your discussion section

Line 350 – 356

This is actually the conclusion, consider reviewing the section

Author Response

REVIEWER #1:

The current study is informative because it informs institutions and healthcare providers on the post- market performance of the drug entity, a parameter not always captured in the clinical studies stage in drug development.

The article is well-written and relevant to the medical practice community. I only have a few comments as follows.

The Authors thank the Reviewer #1 for the overall positive comment to our manuscript and for the comments. We are very pleased that our manuscript has elicited the interest of Reviewer#1.

Q1: The method section - Provide a summary description of the database to include how it can be accessed, what the parameters measure (Variables) what is the year span of the data extracted, and so on.. This provides readers with enough information to understand the database.

A1: We would like to thank Reviewer #1 for this suggestion. The “Material and Method” section has been revised accordingly: “Data was obtained from the FAERS Public Dashboard which is a publicly-available web-based tool that allows for the querying of FAERS data related to ADRs reported to the FDA by the pharmaceutical industry, healthcare providers and consumers. The reports being submitted to the FAERS comprised a range of parameters including: suspect product active ingredient(s), reason for use, seriousness of the ADR, event date, sex of the patient, patient’s age, patient’s weight, reporter type/reporting source, concomitant product(s) taken by the patient at the same time of the report, country where event occurred, possible literature reference(s) where the event was discussed. The whole database was analyzed, including the reports being submitted from 1981 to 2021 (Figure 1).”

Q2: The method section - I think there should be some additions to the analysis. The active agent that is associated with MFI needs to be classified pharmacologically stating in which indication they are used, what are they actually used for the case reported, and maybe to which patient group (Male: age distribution)

Also provide time trends and geological location of the reports, who made the reports, what action was taken, and possibly, the general outcome after an intervention. I hope you understand that all these parameters affect the drug therapy.

Consider that this information might be valuable to patient groups that are not medical experts so more information is needed.

A2: We would like to thank Reviewer #1 for this suggestion. Table 1,2 and 3 have been implemented in the paper, describing the MFI-reporting trends throughout the years, the reporter-type/submitting agent trends and the countries where the MFI-reports have been reported, respectively. Table 1 has been updated to include the pharmacological class of the drugs being associated the most with MFI-reports in the present analysis.

Q3: The conclusion - Comment on Line 345. This is not in your result kindly review. Perhaps suited for your discussion section

A3: We would like to thank Reviewer #1 for this suggestion. As discussed, we identified significant levels of disproportionate reporting for a number of molecules and there is already a body of existing evidence linking some medications with possible drug-induced MFI. It is therefore considered mandatory to collect accurately the drug history of every male patient seeking medical help for fertility issues.

Q4: The conclusion - Line 350 – 356. This is actually the conclusion, consider reviewing the section

A4: We would like to thank Reviewer #1 for this suggestion. The “Conclusions” section has been revised accordingly.

Reviewer 2 Report

Comments and Suggestions for Authors

Although the paper is interesting and valuable, some issues need to be highlighted. Also, correction should be madden order to improve the paper. 

1. Please provide in more details the inclusion/exclusion criteria. Did the authors use for analysis only documents presenting the effects of single drugs on fertility or were publications describing the effects of mixtures of drugs also taken into account?

2. In line with this, I'm wondering whether the patient's health status was taken into account? This should be provided in the methodology section

3. I suggest to rewrite the table by grouping the drugs in specific groups of drugs, e.g. sertraline and fluoxetine as SSRI, valproic acid as antiepileptic, etc.

4.  Instead of summarizing existing knowledge about each drugs (discussion section), I encourage the Authors to emphasize their unique contributions, experimental findings.

5. Please provide at least two diagrams to make the paper interesting and readable. It looks more like a compilation of literature in its present form

Comments on the Quality of English Language

Minor changes/corrections should be provided

Author Response

REVIEWER #2:

Although the paper is interesting and valuable, some issues need to be highlighted. Also, correction should be made in order to improve the paper.

We would like to thank Reviewer #2 for the positive comments and for his/her suggestions.

Q1: Please provide in more details the inclusion/exclusion criteria. Did the authors use for analysis only documents presenting the effects of single drugs on fertility or were publications describing the effects of mixtures of drugs also taken into account?

A1: We would like to thank Reviewer #2 for this welcome request of clarification. Although the FDA reports may include information about any concomitant drugs taken by the patient at the time of the report, this field is often left incomplete by the submitting agent. Moreover, a systematic analysis of the concomitant drugs taken by the patient is difficult to be provided here, as this included a number of medications which are typically not associated with any reports in the database and in the published literature. Therefore, this further sub-analysis has not been included as it may have had hampered the readability of the present paper. However, a mention of this relevant issue has now been included in the Limitations’ section of the paper.

The statement reads as follows: “Furthermore, due to the limitations of the pharmacovigilance databases, only reports presenting the effects of single drugs, as opposed to drug mixtures containing the index drug of concern, were here considered for the analysis”

Q2: In line with this, I'm wondering whether the patient's health status was taken into account? This should be provided in the methodology section.

A2: Many thanks for this insightful suggestion. Indeed, an analysis of the medical history of the patient experiencing the suspect ADR would be of outmost importance to better understand a number of relevant issues, including whether some specific categories of patients may be more prone to develop MFI as a consequence of medication intake, and to clarify better the profile of the patients experiencing this ADR. Unfortunately the pharmacovigilance databases including the FAERS present with a number of issues, including lack of direct data regarding any possible comorbidity of the patients. Indirect data regarding the medical history of the patients can be obtained from the analysis of the concomitant drugs taken by the patient at the time of the report, even though this analysis present with some issues which have been discussed in A1 in response to this Reviewer’s comments. This issue has been discussed in the Limitations Section of the paper as follows: “Indeed, an analysis of the medical history of the patient experiencing the suspect ADR would have been here of outmost importance to better understand a number of relevant issues, including whether some specific categories of patients may be more prone to develop MFI as a consequence of medication intake. However, the patients’ health status is not typically mentioned in the pharmacovigilance databases, including the FAERS”

Q3: I suggest to rewrite the table by grouping the drugs in specific groups of drugs, e.g. sertraline and fluoxetine as SSRI, valproic acid as antiepileptic, etc.

A3: We would like to thank Reviewer #2 for this very welcome suggestion. The table has been revised accordingly.

Q4: Instead of summarizing existing knowledge about each drug (discussion section), I encourage the Authors to emphasize their unique contributions, experimental findings.

A4: We would like to thank Reviewer #2 for this suggestion. The Discussion section of the paper focusses on highlighting the most up-to-date clinical and pre-clinical evidence dealing with the possible influence of those medications which were here associated with significant PRR values. Indeed, the pharmacovigilance approach may present with some clear advantages, which have been discussed throughout the paper, but its main limitations lie with its inability to prove causality. Hence, to help the reader to better interpret current data, we have expanded on the possible physio- pathological mechanisms potentially explaining this hypothetic association between the index drug and MFI occurrence, where appropriate. Consistent with this, rather than describing the pharmacovigilance findings which per se may need to be taken with caution, the Conclusions’ section may better reflect the actual impact of medications on MFI.

Q5: Please provide at least two diagrams to make the paper interesting and readable. It looks more like a compilation of literature in its present form.

A5: The Authors would like to thank the Reviewer 2 for this comment which has given has the occasion to render the present paper more intelligible to the Readers of this Journal

Reviewer 3 Report

Comments and Suggestions for Authors

This paper has a lot of quality and methodological concerns: 

- In the introduction

The authors stated that "a wide range of medications may have a possible role in its pathogenesis" and "the recognised impact which some medications may have in the pathogenesis of MFI, the quality of the evidence in support of drug-induced MFI is relatively low, and the issue remains under-investigated"...what is or what are your rationals? Some of the agents or drug classes you mentioned as possibly related to MFI have already listed this type of event in their labels, so the regulatory authorities have already established the strength and quality of evidence for the association. So your statements to support your study are misleading and not aligned with the knowledge available. 

- lines 68 - 77 You perform a fishing expedition strategy, that is not recommended by the good pharmacovigilance practice guidelines. This is not a high-quality methodology in pharmacovigilance.

- lines 76-77 What is the rationale for this choice? This is not recommended by the good pharmacovigilance practice guidelines. 

- Do you perform PRR only on suspected drugs? Or also on concomitant? As you report in lines 145-147 "mechlorethamine was the 145 sole drug being suspected of being associated with the ADR, while for the remaining 11 146 reports (e.g., 64.7%) there was a mention of vincristine sulfate as well. " I understand that you analyzed also the reports in which the drugs were not reported as suspected, this is not a good methodological approach in pharmacovigilance.

- In lines 150-154 you reported: "This is the first study assessing the possible detrimental impact of medications on the development of MFI using a pharmacovigilance approach" This statement is misleading since regulatory authorities already issued this safety risk for some drugs. Indeed, you found finasteride as the most associated with MFI, note that this event is labeled for this drug in RCP, and in light of your disquisition in the discussion section the evidence is a lot. 

Author Response

REVIEWER #3:

This paper has a lot of quality and methodological concerns.

Q1: In the introduction. The authors stated that "a wide range of medications may have a possible role in its pathogenesis" and "the recognised impact which some medications may have in the pathogenesis of MFI, the quality of the evidence in support of drug-induced MFI is relatively low, and the issue remains under- investigated"...what is or what are your rationals? Some of the agents or drug classes you mentioned as possibly related to MFI have already listed this type of event in their labels, so the regulatory authorities have already established the strength and quality of evidence for the association. So your statements to support your study are misleading and not aligned with the knowledge available.

A1: Many thanks for this insightful comment. MFI is a prevalent and increasing issue in the general population, and male patients presenting to the fertility clinic on long-term medications are likely on the increase. Even though we may agree that for a number of medications which we identified here as being associated with significant disproportional reporting levels there is already some scientific evidence of a possible impact on MFI, the issue still remains under-investigated in the literature. Indeed, a number of chemotherapy agents have already been associated with robust levels of evidence relating to their influence on spermatogenesis. Similarly, there are very convincing literature suggestions linking testosterone intake and MFI, although some issues of relevance, such as the reversibility of the effect of testosterone on spermatogenesis, still need to be better clarified. Conversely, most of the clinical studies dealing with the possible impact of 5ARIs on male fertility present with methodological concerns and conflicting findings, and this possible association with MFI is still considered a controversial topic in the literature.

Finally, for a number of molecules here identified (e.g. antiepileptics, antidepressants, beta-blockers) the evidence in support of their association with MFI is extremely scant, mainly because the issue has not been investigated enough, despite the existence of a scientific rationale to better investigate on this association. We primarily aimed here at better clarifying which medications may deserve more clinical research attention to better clarify their association with MFI. Finally, even though for some of these drugs it has already been issued a warning by the Regulatory Agencies, these warnings remain scientifically debatable, and the sources of these warnings still lie with the integration of the published clinical studies and the analysis of the post- marketing findings. Consistent with the Reviewer’s advice, a statement in the Introduction have been modified as follows: “Despite for some molecules, such as chemotherapy agents there are convincing levels of evidence relating to their association with MFI (6), for other molecules the quality of the evidence in support of this drug-induced association is either conflicting or relatively low/unsatisfactory, despite the existence of a scientific rationale to better investigate on this issue”

Q2: lines 68 - 77 You perform a fishing expedition strategy, that is not recommended by the good pharmacovigilance practice guidelines. This is not a high-quality methodology in pharmacovigilance.

A2: Many thanks for this comment. The computation of the PRR values and their CI levels is still considered the golden standard strategy to identify any possible signal of disproportionate reporting for any given medication. Consistent with this, we have included the following statement in the Limitations’ section: “One could wonder about the pharmacovigilance consistency of carrying out indeterminate searches (at times referred to as “fishing expedition”) in order to obtain, as the case may be here, information relating to the association between MFI and an index drug. Conversely, the computation of the PRR values and their CI levels is still considered the golden standard strategy to identify any possible signal of disproportionate reporting for any given medication”

Q3: lines 76-77 What is the rationale for this choice? This is not recommended by the good pharmacovigilance practice guidelines.

A3: We would like to thank Reviewer #3 for this comment.

Consistent with your advice, we have included the following statement in the Limitations’ section: “One could wonder about a possible pharmacovigilance bias here introduced in assessing only those drugs being associated with more than 10 MFI reports each. This has been carried out to avoid the over-inclusion in the paper of large numbers of mainly anecdotal, and likely not to reach the statistical significance, reports relating to single molecules. A similar approach has been implemented as well in previous pharmacovigilance studies (73)”

Q4: Do you perform PRR only on suspected drugs? Or also on concomitant? As you report in lines 145-147 "mechlorethamine was the 145 sole drug being suspected of being associated with the ADR, while for the remaining 11 146 reports (e.g., 64.7%) there was a mention of vincristine sulfate as well. " I understand that you analyzed also the reports in which the drugs were not reported as suspected, this is not a good methodological approach in pharmacovigilance.

A4: Many thanks for this needed request of clarification.

Consistent with both your advice and Reviewer’s #2 comments, we have included the following statement in the Limitations’ section: “Although the concomitant drugs taken by the patient at the same time of the suspected drug were here, whenever possible, disclosed, the PRR values were computed only for the suspected drugs and not for the molecules being identified in the drug combination, and this may make more difficult the interpretation of current data”.

Q5: In lines 150-154 you reported: "This is the first study assessing the possible detrimental impact of medications on the development of MFI using a pharmacovigilance approach" This statement is misleading since regulatory authorities already issued this safety risk for some drugs. Indeed, you found finasteride as the most associated with MFI, note that this event is labeled for this drug in RCP, and in light of your disquisition in the discussion section the evidence is a lot.

A5: We would like to thank Reviewer #3 for this comment. We respectfully disagree with Reviewer’s #3 comments on this point (see also A1 in response to this Reviewer’s comment). Indeed we recognize that the 5ARIs are already recognized to have a detrimental effect on spermatogenesis on some patients, as safety warnings have already been issued. Although the implications of finasteride-association with MFI are of outmost clinical importance, the evidence in support of the possible impact of finasteride on male fertility levels is however relatively scant and conflicting, and some issues of relevance such as which patients are more susceptible of this effect (e.g., effect on spermatogenesis on the fertile vs. the unfertile patients) still remain a matter of debate. Although the Regulatory Agencies are clearly well aware of the signals of disproportional reporting linking the 5ARIs with MFI, to the very best of our understanding the present study would be the first to disclose to the scientific community which drugs are associated the most with disproportionate reporting levels for MFI.

Consistent with your advice, the Discussion section statement has been modified as follows: “This study adds to the existing, albeit for some medications conflicting or unsatisfactory, evidence of the possible detrimental impact of medications on the development of MFI using a pharmacovigilance approach”

Round 2

Reviewer 1 Report

Comments and Suggestions for Authors

Thanks very much for responding to the comments raised in the first rounds of review. I have a few more comments and questions to improve the content of the manuscript.

I think all the figures are not well presented and need to be looked into.

Figure 1: there are no labels on the x and y-axis please revise

Figure 2: The y-axis has no labels; please revise. Review accordingly

Figure 3: Label the axis and define the abbreviation of the countries below the table [eg. Australia AU]

Figure 4: There is no percentage; I am just seeing the two colors

Comments on the Quality of English Language

The English is comprehensible; maybe a few corrections are needed

Author Response

REVIEWER #1:

Thanks very much for responding to the comments raised in the first rounds of review. I have a few more comments and questions to improve the content of the manuscript.

Q1: I think all the figures are not well presented and need to be looked into.

Figure 1: there are no labels on the x and y-axis please revise

Figure 2: The y-axis has no labels; please revise. Review accordingly

Figure 3: Label the axis and define the abbreviation of the countries below the table [eg. Australia AU]

Figure 4: There is no percentage; I am just seeing the two colors

A1: We would like to thank Reviewer #1 for these comments. All of the figures were revised accordingly.

Reviewer 2 Report

Comments and Suggestions for Authors

The Authors improved the paper greatly, Therefore, in my opinion it is suitable for further publication. Congratulations!

Author Response

REVIEWER #2:

The Authors improved the paper greatly, Therefore, in my opinion it is suitable for further publication. Congratulations!

We would like to thank Reviewer #2 for the positive comment.

Reviewer 3 Report

Comments and Suggestions for Authors

Thank you to the authors for their attempt to improve the manuscript. However, I want to notice you that to insert in limitations the methodological concerns, is not a good attempt for improving when the methodological issues are so many. Even though I appreciate your effort to try to give quality to your work. Consequently, I want to give you a chance. There are three main methodological issues that cannot be leave to the case. 

If you will solve these three major deviations, by re-performing the analyses in accordance I believe that your work could reach the standard of quality needed by the pharmacovigilance studies. Here below the mandatory analyses to be performed:

- First, you must to re-perform all the analyses by considering the masking effect of drugs for which the MFI is labelled in RCP. So you must carried out the PRR by considering the unmasking for the remaining suspected drugs. Use the example by Capogrosso Sansone A., et al 2017 [Drug Saf. 2017 Oct;40(10):895-909.doi: 10.1007/s40264-017-0564-8].

- Second, yes all the analyses have to be performed only on the suspected drugs but the evaluation of concomitant medications cannot be left a part. So please, you have to integrate with this evaluation of the concomitant medications. As you know, concomitant medications can have an effect in sense of adding or decreasing the ADR detected. 

- Third, a quality evaluation about the causality of suspected drugs analyzed is also required. Please, you have to add this important evaluation. 

Author Response

REVIEWER #3:

Thank you to the authors for their attempt to improve the manuscript. However, I want to notice you that to insert in limitations the methodological concerns, is not a good attempt for improving when the methodological issues are so many. Even though I appreciate your effort to try to give quality to your work. Consequently, I want to give you a chance. There are three main methodological issues that cannot be leave to the case. 

If you will solve these three major deviations, by re-performing the analyses in accordance I believe that your work could reach the standard of quality needed by the pharmacovigilance studies. Here below the mandatory analyses to be performed:

Q1: First, you must to re-perform all the analyses by considering the masking effect of drugs for which the MFI is labelled in RCP. So you must carried out the PRR by considering the unmasking for the remaining suspected drugs. Use the example by Capogrosso Sansone A., et al 2017 [Drug Saf. 2017 Oct;40(10):895-909.doi: 10.1007/s40264-017-0564-8].

Q2: Second, yes all the analyses have to be performed only on the suspected drugs but the evaluation of concomitant medications cannot be left a part. So please, you have to integrate with this evaluation of the concomitant medications. As you know, concomitant medications can have an effect in sense of adding or decreasing the ADR detected. 

A1/2: Many thanks for these insightful comments and welcome suggestions. The provided literature reference has been thoroughly evaluated. A number of issues of relevance should be raised to this respect. The Capogrosso Sansone et al study assessed the PPI-associated reporting odds ratios (RORs) and their 95% confidence intervals (CIs) as a measure of disproportionality for muscular ADRs in an Italian pharmacovigilance database. In a secondary subanalysis, Capogrosso Sansone et al analyzed the same issue after taking into account a potential masking effect of statins. More in detail, the Authors performed an “unmasking” analysis by repeating the primary analysis after excluding from the dataset all reports in which at least one statin was suspected of being the causative drug. This approach may be feasible and may provide potentially useful findings when a single widely used class of medications (or very few ones) are frequently co-administered with any index drug which is already very well known for being associated with the ADR of interest (i.e. statins and PPIs in the Capogrosso Sansone et al study). Moreover the “unmasking” analysis in the Capogrosso Sansone et al study was performed with the specific purpose to investigate whether there could be a masking effect relating only to statins. In their analysis, in fact, the Authors were very well aware that statins are associated with a very significant number of muscular-damage effects reports. Performing the “unmasking” analysis was certainly appropriate for their study, even though one could argue that this choice might have been arbitrary as the same “unmasking” analysis was not performed for every single medication which is well known as well for causing muscular damage such as antithyroid drugs, fibrates, cyclosporine, colchicine, antiretrovirals, chemotherapy agents, corticosteroids etc.

Hence, in line with the Reviewer no 2 advice, in the Materials and Methods section the following statement has been included: ‘…Consistent with Capogrosso Sansone et al (9), a range of secondary, ‘unmasking’, analyses were here carried out. The identification of candidates masking products still however relies on an empirical approach (10). Hence, to be as conservative as possible, unmasking was here obtained in repeating the primary analysis whilst excluding from the dataset relating to drugs associated with a significant PRR value all reports in which at least one other drug (e.g. any drug, irrespective of having been suspected of being associated with MFI) was mentioned as a concomitant medication……’. Furthermore, in the Results’ section the PRR values resulting from the secondary, unmasking, analyses were now shown as well.

In the Discussion section, we have also added the following statement: ‘….Following the unmasking exercise, it appeared from here that some signals of disproportionate analysis actually increased in value, whilst others resulted to be non-significant, with this effect having been described before (9). Although the highest masking effects are obtained by products for which the reaction is known or has been extensively reported (10). a conservative approach was here taken into account and any combination drug, irrespective of having been the subject of an MFI concern or not, was eliminated from the related dataset of interest. One could argue that this may have further amplified the differences between the ‘crude’ and ‘unmasked’ PRR values of interest. …’

Regarding Q2, the following statement has been included in the Limitations’ section: ‘…..Although the concomitant drugs taken by the patient at the same time of the suspected drug were here, whenever possible, disclosed, the PRR values were computed only for the suspected drugs and not for the molecules being identified in the drug combination. On the other hand, only procarbazine and vincristine, both associated with mechloretamine, were here identified in more than 3 instances out of the whole dataset and with the related unmasking analysis their confounding/masking role on mechloretamine was here possibly better interpreted….’

Q3: Third, a quality evaluation about the causality of suspected drugs analyzed is also required. Please, you have to add this important evaluation.

A3: Many thanks for this insightful comment. Consistent with your advice, in the Methods’ section we have included the following statement: ‘…….To carry out a quality evaluation about the causality of the suspected drugs analyzed, the Naranjo probability scale (11) was here used. Consistent with Gupta and Kumar (12) the scores were independently calculated by two clinicians (e.g. a clinical pharmacologist and a urologist; FS and NS) and possible disagreement issues were discussed ….’. In the Results’ section, the following statement was added: ‘… According to the Naranjo probability scale, all suspected drugs here associated with significant levels of disproportionate reporting scored as having had a ‘possible’ role in causing the MFI ADR…..’

Finally, we have included in the Limitations’ section the following statement: ‘….Although among the many tools identified, the Naranjo's scale remains the most comprehensive and easy to use for performing causality assessment of ADRs (79), its applicability in the current study was of limited value. In fact, a number of clinical information to optimally compile the scale (e.g.: reappearance of the ADR when a placebo was given; presence of toxicity levels of the suspected drug; occurrence of the same ADR reactions with previously administered drugs etc) were not here made available in the related reports. It is an inherent limitation of spontaneous reporting, however, that firm evidence cannot usually be produced.…..’

Round 3

Reviewer 3 Report

Comments and Suggestions for Authors

Congratulations and thank to the authors for improving their work that now has achieved the standards of quality as regards the methodological requirement for pharmacovigilance studies! A good research is that ensures transparency and reproducibility. I would appreciate also to find in the manuscript a table providing the results of the secondary analyses to make reader know how has been changed the distribution of the reports associated with the pair drug-MFI under the unmasking effect. Please, include this.

Author Response

Congratulations and thank to the authors for improving their work that now has achieved the standards of quality as regards the methodological requirement for pharmacovigilance studies! A good research is that ensures transparency and reproducibility.

A0: Many thanks for the positive comment

Q1:I would appreciate also to find in the manuscript a table providing the results of the secondary analyses to make reader know how has been changed the distribution of the reports associated with the pair drug-MFI under the unmasking effect. Please, include this.

A2: Many thanks for the welcome suggestion. Table 2 dealing with the secondary analyses being performed has been included in the manuscript.